# Alpine Viticulture and Climate Change: Environmental Resources and Limitations for Grapevine Ripening in Valtellina, Italy

**DOI:** 10.3390/plants12112068

**Published:** 2023-05-23

**Authors:** Davide Modina, Gabriele Cola, Davide Bianchi, Martino Bolognini, Sonia Mancini, Ivano Foianini, Adriano Cappelletti, Osvaldo Failla, Lucio Brancadoro

**Affiliations:** 1Department of Agricultural and Environmental Sciences, University of Milan, 20133 Milano, Italy; davide.modina@unimi.it (D.M.); davide.bianchi3@unimi.it (D.B.); martino.bolognini@unimi.it (M.B.); acappelletti@fondazionefojanini.it (A.C.); osvaldo.failla@unimi.it (O.F.); lucio.brancadoro@unimi.it (L.B.); 2Fondazione Foanini di Studi Superiori, 23100 Sondrio, Italy; presidente@fondazionefojanini.it (S.M.); ifoianini@fondazionefojanini.it (I.F.)

**Keywords:** alpine viticulture, mountain viticulture, ripening, acidity, sugar content, environmental resources and limitations, heatwaves, climate change

## Abstract

The effects of the spatial and temporal variability of environmental factors on viticulture are particularly important in mountainous wine regions due to their complex geomorphology. A typical example is Valtellina, an Italian valley in the middle of the Alpine chain known for its wine production. The aim of this work was to assess the effects of the current climatic conditions on Alpine viticultural production by evaluating the relationship between sugar accumulation, acid degradation, and environmental factors. To achieve this objective, a 21-year time series of ripening curves from 15 vineyards (cv Nebbiolo) along the Valtellina wine-growing belt was collected. The ripening curves were then analysed in conjunction with meteorological data to assess the influence of geographical and climatic characteristics, as well as other limiting environmental factors, on grape ripening. Valtellina is currently characterised by a stable warm phase, with yearly precipitation slightly higher than in the past. In this context, the timing of ripening and the level of total acidity are correlated with altitude, temperature, and summer thermal excess. Precipitation shows good correlations with all the maturity indices, so higher precipitation leads to late ripening and higher total acidity. Considering the oenological goal of local wineries, the results suggest that the Alpine area of Valtellina is currently facing favourable environmental conditions, with early development and increased levels of sugar while maintaining good levels of acidity.

## 1. Introduction

Climate plays a crucial role in viticulture [1,2,3] and is fundamental in defining the viticultural potential of an area [4]. This is because various meteorological variables, such as solar radiation, temperature, precipitation, relative humidity, and wind, exert influence on vine development and grape ripening [2].

The environmental driving variables are characterised by spatial and temporal variability. Space variability contributes to defining the terroir of a wine region by affecting the vocation of the area toward wine production. Time variability determines the vintage effect due to the peculiar behaviour of meteorological conditions during the growing season and ripening process [5]. When considering spatial variability, three distinct climate categories can be identified: macro-climate, meso-climate, and micro-climate. The macroclimate refers to broad regional areas, mostly defined by the latitude and longitude of a wine region. The meso-climate takes into account local variations in geographical factors such as altitude, slope, exposure, water basins, etc. Lastly, the micro-climate focuses on variability observed at the level of individual fields, plant canopies, and even single organs [6]. 

In mountainous wine regions, the analysis of meso-climate and its interaction with grapes is particularly important. These areas are usually characterised by diverse geomorphological features, including significant variations in altitude, aspect, slope, and soil types. These factors collectively contribute to the establishment of a wide range of zones with varying potential for grape quality. Several studies highlight different sugar and acid concentrations in grapes from vineyards at different altitudes [7,8,9]. These variations in grape characteristics are commonly attributed to differences in the radiative and thermal regimes, which are primarily influenced by altitude in mountainous regions (with a general assumption of temperature decreasing by 0.45–0.65 °C per 100 m altitude, depending on the location, time of year, and air circulation), as well as by aspect [10,11]. 

Solar radiation is the main environmental factor for plant development, being the source of energy for photosynthesis and the main driver of the evapotranspiration process [11,12]. During the ripening process, these environmental factors also impact the quality of grapes, influencing parameters such as sugar concentration, total acidity, and pH of the berries [13], as well as their phenolic composition [14]. Furthermore, the combination of high radiative input, high temperatures, and UV radiation can lead to sunburn, which involves the accumulation and oxidation of phenolics compounds [15,16,17]. This impacts the quality of production [18] through cracking, berry desiccation [18,19], and poor colour development of red varieties [20].

Temperature is a fundamental factor in vine development, affecting photosynthetic activity [21] and phenological timing. During grape ripening, high temperatures promote a decrease in total titratable acidity [22]. Furthermore, the relationship between temperature and grape sugar concentration exhibits a curvilinear pattern, with the highest levels typically occurring at temperatures ranging between 25 °C and 30 °C [23]. In addition, the accumulation of anthocyanins in grapes shows an optimal range of temperatures between 17–26 °C [19,24]. Cool night temperatures have been found to facilitate sugar accumulation, limit vegetative growth [25,26], enhance the coloration of red grapes [27], and generally contribute to the favourable aromatic expression of wines [4]. During grape ripening, temperatures also affect the concentrations of potassium and proline, which increase linearly with temperature while the concentration of malate decreases [4]. Variation in temperatures also promotes different characteristics in wine from the same variety. For instance, Jones [3] reported that Pinot Noir wines in the coolest climates are lighter and more elegant, while the warmer zones produce fuller-bodied, fruit-driven wines. High temperatures, often related to extreme events such as heat waves, play a relevant negative role since they could inhibit photosynthesis [21] and influence the ripening process. In this regard, a delay in phenolic maturity and a reduction in colour development due to lower anthocyanin production are also associated with the occurrence of hot days during ripening [28]. 

Rainfall quantity and distribution, mediated by the vineyard soil hydraulic properties, determine soil water availability and, therefore, the plant’s water status [2]. This, in turn, affects plant production in terms of quantity and quality, playing a fundamental role in terroir expression [4]. Depending on the phenological stage, vine water status has a different impact on grape quality [29]. Particularly, high water availability during ripening time promotes more vegetative growth and reduces sugar, colour, flavour levels, and phenol concentrations in berries [30]. On the other hand, severe water shortages can have a negative impact on vine growth and fruit development, negatively affecting proper ripening [31].

Several studies account for climate change’s present and future effects in relevant viticultural areas at different spatial scales [32,33,34,35]. In the specific case of mountain viticulture, some studies already report the effects of climate change in wine regions located in the Italian Alps, particularly in the Trentino-Alto Adige region. The recent increase in temperature [36] has driven a significant advance in grapevine phenological stages both on the valley floor and in mountain locations [37], and a similar trend is expected in the following years. In this context, the use of higher-altitude vineyards could be a promising option [38,39,40,41] for adaptation to the new climatic conditions. For instance, recent plantings in the Alpine region of Alto Adige are more frequently located at higher altitudes than in the past [40]. Furthermore, the effects of temperature increases are anticipated to be more pronounced at higher altitudes [42]. Although significant deviations from current conditions may not be expected in the near future, certain mountainous regions situated around 1000 m in altitude could potentially become suitable for viticulture by the end of the century [37]. 

Valtellina is an Alpine, east–west oriented valley located in the Northern part of the Lombardy region in Italy. It is the upper valley of the Adda River, a main left tributary of the Po River, and has been renowned for its wine production since Roman times. The dominant vine in the area is the late-ripening cv Nebbiolo, which aligns with the main objective of producing red wines with robust ageing potential. This makes the analysis of the local viticultural model particularly interesting, especially in relation to current and future climate changes. 

Approximately 20 years ago, a viticultural zoning study of the area was conducted, and the results, reported by Failla et al. [43], showed that altitude and yearly potential photosynthetically active radiation (PPAR) have an impact on the timing of bud break and flowering. Additionally, the date of veraison is influenced by PPAR and its interaction with the vine crop load. Technological maturity is mostly influenced by altitude, while phenolic maturity is influenced by crop load, PPAR, and its interaction with crop load and altitude. The highest phenolic maturity was recorded in low-altitude and low-PPAR vineyards.

Over the past 21 years, the research and technical assistance centre of Fondazione Fojanini has constantly monitored the progress of ripening in 15 commercial vineyards, representative of the Valley wine areas. The aim of this study was to analyse the time series of ripening curves and meteorological data of the 15 vineyards to assess the relationship between sugar accumulation, acidic degradation, and environmental factors. 

## 2. Results and Discussion

### 2.1. Environmental Conditions in Valtellina

This study evaluates data collected by the technical assistance centre of Fondazione Fojanini in 15 commercial vineyards (V1 to V15), representative of the viticultural belt along the Rhaetian mountainside of the valley (Figure 1).

Figure 2 shows the last 41 years of data for the weather station of Sondrio from the network of ARPA Lombardia (the Regional Environmental Agency of Lombardy), in the middle of the wine belt of Valtellina, close to vineyards V8–V11.

The yearly average temperature course follows the typical behaviour of Western Europe [44]. It experienced a gradual rise, starting in the late 1980s and reaching a stable level in the early 2000s, marking the onset of the current warm phase. To account for this change, the 20-year climate normal of 1981–2000 and 2001–2020 have been chosen as references instead of the 30-year standard suggested by the WMO (2022).

The increase in temperatures had obvious effects on the occurrence of heat waves (Figure 2), which moved from an average yearly value of 13 d y^−1^ during the past climate phase to 21 d y^−1^ in the current one. 

Regarding precipitation, the two phases show a yearly average value that increased from 876 mm in the 1981–2000 period to a 2001–2020 value of 1013 mm (+15%), with increased interannual variability (Figure 2).

Geographical and climatic features of the vineyards.

Based on DTM and meteorological data, the following geographical and climatic features (Table 1) were derived for the 15 vineyards: Elevation [m]Aspect [°]PPAR—Potential Photosynthetically Active Radiation [MJ]ATY—2001–2020 average of yearly mean temperature Tmean [°C]PrecY—2001–2020 accumulation of yearly precipitation [mm]HWY—2001–2020 sum of yearly days with maximum daily temperature Tmax above 32 °C [n days]

Most of the vineyards’ aspect ranges from south–east to south–west with PPAR values in the 3300 to 3500 MJ/m^2^ range. The only exception is V14, East-exposed, with the lowest PPAR of 2794 MJ/m^2^. Yearly average temperature (ATY) ranges between 12.1 °C (V7) and 14.2 °C (V5 and V2), while yearly precipitation (PrecY) is between 1022 mm (V6) and 1255 mm (V13). The most noticeable difference among the vineyards is provided by the yearly number of heat waves (HWY), from 15 in V14 to 38 in V5.

### 2.2. The Ripening Progress

The first analysis performed focuses on the ripening indices (Table 2).

Considering DOY15 (the day of achievement of the grape sugar content of 15 °Brix, assumed as an early ripening stage), the average value is 237 (25 August) with a standard deviation of 8.9 days. The earliest value is 221 (9 August) and the latest is 254 (11 September).

In the case of DOY20 (the day of achievement of the grape sugar content of 20 °Brix, representing technical maturity), the average value is 259 (16 September) with a standard deviation of 9.6 days. The earliest value was 235 (22 August), and the latest was 287 (10 October).

Ripening Length (RL—as the difference between DOY20 and DOY15) is on average equal to 22 days, with a standard deviation of 4.7 days. The shortest period is 14 days, and the longest is 33.

Total acidity at technical maturity (TA) is, on average, equal to 11.01 g L^−1^ with a standard deviation of 1.66 g L^−1^, with the lowest value of 7.95 g L^−1^ and the highest value of 14.33 g L^−1^.

The variability of each vineyard for 21 years and each year for 15 vineyards is analysed using box plots, as shown in Figure 3 and Figure 4.

Regarding the vineyard variability (Figure 3), the vineyards show consistent behaviour across all four indices examined. There is a strong correlation between the average values of DOY15 and DOY20 (R^2^ = 0.85), as well as between average DOY15 and average TA (R^2^ = 0.52). However, the relationship between DOY20 and TA displays less consistency. 

The vineyard variability of DOY15 is quite similar in all the vineyards, with V4 and V10 being the ones that exhibit a higher variation. Similarly, in the case of DOY20, the highest variability is shown by V2, V4, V10, V13, and V15. The ripening length is particularly stable among the vineyards, except for V10 and V14, which exhibit a longer duration. 

In the case of V10, the vineyard shows average late DOY15 and DOY20. V10 is the second highest for elevation among the studied vineyards (613 m a.s.l.), and the late and longer ripening can be explained by a lower thermal regime (12.7 °C of yearly average temperature). 

In the case of V14, the vineyard position on the South–Eastern side of the Valley determines a lower thermal regime despite the lower altitude (474 m a.s.l. and 12.9 °C of average yearly temperature). 

V15, the highest in altitude at 623 m, is characterised by more favourable thermal conditions, with a yearly average temperature of 13.1 °C. V15 shows a late average DOY15, but this is not translated into a late DOY20 or a longer RL.

In the case of seasonal variability (Figure 4), the four indices are again consistent, with significant correlations between DOY15 and DOY20, DOY15 and TA, and DOY2 and TA with R^2^ values of 0.88, 0.48, and 0.37, respectively. For DOY15 (the average day of the 15 vineyards), the earliest seasons are 2003, 2007, 2009, 2011, 2017, and 2018, with days of occurrence between 12 and 18 August. The latest seasons are 2001, 2002, 2008, 2013, and 2016, all dropping in the first week of September. 

For DOY20, the earliest average dates are in 2003, 2007, 2009, 2011, 2017, and 2018 and are comprised of between the end of August and the first 10 days of September. The latest seasons are 2001, 2008, 2013, 2014, and 2016, all in the last week of September.

In the case of total acidity, values below 10 g L^−1^ were obtained in 2003, 2011, 2015, 2017, and 2020, most of which were years of early ripening (DOY15 and DOY20). Values of TA above 12 g L^−1^ were obtained in 2001, 2002, 2008, and 2014, all seasons with late DOY20. 

Results highlighted that viticultural activity in Valtellina is strongly affected by the interannual variability of environmental conditions [45,46,47]. Additionally, this should be considered to define effective agronomical management of the vineyard [48,49].

To assess a possible trend in the timing of ripening during the period of study, linear regressions between year and DOY15, DOY20, RL, and TA were set up (Table 3). 

Even with very low values of R^2^, significant advances in DOY20 (−0.345 d y^−1^), decreases in TA (−0.037 g L^−1^ y^−1^), and shortenings in RL (−0.107 d y^−1^) were found when considering the whole ensemble of data from the 15 vineyards. This trend agrees with similar results detected in other wine regions in recent years [50]. However, analysing the data separately for each vineyard, no significant regression between year and DOY15 or between year and TA was found. A significant advance in DOY20 was found only in 3 vineyards (V6 R^2^ = 0.148, −0.571 d y^−1^; V10 R^2^ = 0.187, −0.675 d y^−1^; and V13 R^2^ = 0.187, −0.686 d y^−1^), while a significant shortening of ripening length was found only in 2 (V6 R^2^ = 0.277, −0.464 d y^−1^; V16 R^2^ = 0.316, −0.401 d y^−1^).

### 2.3. Ripening and Environmental Conditions

#### 2.3.1. Influence of Geographical and Climatic Features

For each vineyard, a linear regression was set up between the values of DOY20, DOY15, RL, and TA averaged for the 2001–2021 period and the geographical and climatic features (Elevation, Aspect, PPAR, PrecY, ATY, and HWY). Figure 5 shows only the significant regressions.

Elevation shows a high significant (*p*-value ≤ 0,001) positive regression with DOY15 (R^2^ = 0.717; +0.028 d m^−1^) and TA (R^2^ = 0.608; +0.0073 g L^−1^ m^−1^) and at a lower level (0.01 ≥ *p* value > 0.001) with DOY20 (R^2^ = 0.554; +0.0264 d m^−1^). In other words, as elevation increases, the ripening process tends to occur later in the season. Additionally, at the point of technical maturity, the TA of the grapes tends to be higher.

The yearly average air temperature (ATY) has a highly significant (*p*-value ≤ 0,001) negative regression with DOY15 (R^2^ = 0.731; −5.304 d °C^−1^) and DOY20 (R^2^ = 0.709; −5.6081 d °C^−1^) and at a lower level (0.01 ≥ *p*-value > 0,001) with TA (R^2^ = 0.456; −1.1891 g L^−1^ °C^−1^). In this case, with increasing temperatures, ripening occurs earlier in the season while TA at technical maturity is lower.

Air temperature is strongly related to altitude and, to a lesser extent, to PPAR and exposure, resulting from the surface energy balance [11]. In light of this, a strict relation between altitude and ripening was generally found [7,8,9], and in Valtellina, Failla et al. [43] reported that elevation mainly affected technological maturity.

The number of heatwaves HWY showed a significant negative relationship with DOY20 (R^2^ = 0.3415; −0.3415 d d^−1^). This means that hot conditions determine early technical maturity. 

RL does not show any significant regression with the geographical and climatic indices. A possible reason is that inter-year variability is prevalent compared to the geographical and climatic components.

Furthermore, no significant regression was found between PPAR, aspect, YTP and DOY15, DOY20, and TA. This is probably due to the homogeneity of Aspect and PPAR along most of the vineyards. While the role of PPAR and Aspect are partially covered by the ATY index, the lack of correlation with PrecY can be attributed to the fact that in the Valtellina region, where annual precipitation is usually sufficient for the growth requirements of grapevines, the timing of grapevine events is more influenced by the patterns and dynamics of precipitation events [28,51].

#### 2.3.2. Ripening and Environmental Limiting Factors

The four ripening indices were put in relation to environmental indicators through linear regression, considering the whole body of data and the single vineyards. 

The environmental indices adopted are:HHH15—accumulation of high heat hours (representing thermal excess) from the beginning of the season to the day of achievement of 15 °Brix.HHH20—accumulation of high heat hours from the beginning of the season to the day of achievement of 20 °Brix.ΔHHH—accumulation of high heat hours from the day of achievement of 15 °Brix to the day of achievement of 20 °Brix.HW15—number of heat waves from the beginning of the season to the day of achievement of 15 °Brix.HW20—number of heat waves from the beginning of the season to the day of achievement of 20 °Brix.ΔHW—number of heat waves from the day of achievement of 15 °Brix to the day of achievement of 20 °Brix.Prec15—total precipitation from the beginning of the season up to the day of achievement of 15 °Brix.Prec20—total precipitation from the beginning of the season up to the day of achievement of 20 °Brix.ΔPrec—total precipitation from the day of achievement of 15 °Brix to the day of achievement of 20 °Brix.

The results of the regressions are reported in Table 4, Table 5, Table 6 and Table 7. 

When considering all the vineyards collectively, it is observed that DOY15 shows a significant positive regression with Prec15 (R^2^ = 0.237, +0.021 d mm^−1^), as well as a significant negative regression with HHH15 (−0.019 d °C^−1^) and HW15 (−0.482 d d^−1^). However, the R^2^ values for the latter two variables, HHH15 and HW15, are relatively low (0.078 and 0.089 respectively), with the significance of the regression analysis being enhanced by the large dataset used for analysis.

At the single vineyard level, it is observed that Prec15 shows significant regression in 11 vineyards (R^2^ from 0.217 to 0.516). On the other hand, HHH15 shows significant regression in only one vineyard (V11 with R^2^ = 0.216), and HW15 shows significant regression in five vineyards (R^2^ from 0.212 to 0.260). 

It is interesting to note that with the exclusion of V11, the significant negative effect of HW15 is related to the vineyards at the highest altitudes (V6, V7, V10, and V15), where thermal resources are lower. The results indicate that, in general, DOY15 is more influenced by Prec15 (cumulated precipitation) than by thermal excess. Higher levels of cumulated precipitation are associated with an earlier start of maturation (DOY15). This is probably caused by a higher plant vigour related to a better plant water status [52].

DOY20, considering the whole dataset, shows significant negative regression with ΔHHH (R^2^ = 0.36, −0.16 d °C^−1^) and ΔHW (R^2^ = 0.25, −1.26 d d^−1^) and positive regression with Prec20 (R^2^ = 0.21, +0.02 d mm^−1^). Significant negative regressions with HHH20 (−0.03 d °C^−1^) and HW20 (−0.22 d d^−1^) were found as well, but characterised by lower R^2^ (0.15 and 0.13, respectively). Finally, a positive regression, but at a lower significant level (0.05 > *p* value > 0.01), was found with Δprec (R^2^ = 0.02, +0.03 d mm^−1^). 

At the vineyard level, ΔHHH, ΔHW, and Prec20 have significant regression in most of the vineyards (14, 9, and 11, respectively). The result shows that DOY20 is generally more affected by thermal excesses that occur during ripening, causing an advance in the achievement of 20° brix. In the case of precipitation, higher regressions are obtained when considering the total amount of cumulated precipitation (Prec20) instead of the portion cumulated during ripening (Δprec). This leads to the idea that the timing of ripening is more related to the plant’s water status throughout the whole growing season.

Considering the whole data set, TA shows significant negative regression with ΔHW (R^2^ = 0.51, −0.24 g L^−1^ d^−1^), ΔHHH (R^2^ = 0.34, −003 g L^−1^ °C^−1^), HHH20 (R^2^ = 0.23. −0.01 g L^−1^ °C^−1^), and HW (R2 = 0.21. −0.05 d d^−1^). A positive and significant regression was found between Prec20 and the parameter under consideration (+0.01 g L^−1^ mm^−1^). However, the R^2^ value for this regression was relatively low, at 0.10.

Analysing the vineyard-specific results, TA is more linked to thermal excesses during ripening. ΔHW and ΔHHH have the highest R^2^, and their regression is significant in most vineyards (10 and 15, respectively). Indeed, high temperatures during ripening could increase the rate of acid degradation in berries [22]. A positive influence of Prec20 is also observed, even if weak. This is probably linked to acid accumulation at the beginning of the ripening process since the regression between TA and Δprec shows lower significance (0.05 ≤ *p* value > 0.01) and R^2^ (0.02).

Considering the whole dataset, RL shows a significant positive regression with Δprec (R^2^ = 0.18. 0.05 d mm^−1^) and a less significant but still present regression with ΔHHH (R^2^ = 0.03. 0.03 day °C^−1^) and ΔHW (R^2^ = 0.03. 0.21 d d^−1^). Focusing on the singular vineyards, ΔHHH, and ΔHW shows a significant positive regression in very few vineyards (3 and 1, respectively) with low R^2^ value. In six vineyards, Δprec has a significant positive regression. Thermal excess during ripening may, in some cases, have a negative influence on the physiology of maturation, as indicated by the modest correlations found with RL. However, the relationships observed in this case are not very strong. Indeed, Δprec has the largest influence on RL. This could be related to the phenomenon of sugar dilution in berries related to precipitation that occurs during ripening time [28]. 

In conclusion, the increase in Prec induces a higher TA and a delay in reaching DOY15 and DOY20, while the increase in Δprec is related only to higher values of RL. Conversely, thermal excesses that occur during ripening (ΔHHH and ΔHW) time have more influence on DOY20 and TA than the total amount (HHH and HW). Indeed, ripening period temperatures are important for quality wine production [53]. Finally, thermal excesses show a lower effect on DOY15 and RL than precipitation.

#### 2.3.3. Modelling the Timing of Ripening

The predictive models for the achievement of 15 and 20 °Brix are based on the accumulation of NHH (normal heat hours, representing hourly thermal resources useful for grapevine development) and GDD (growing degree days with a base temperature of 10 °C) on the day when the two sugar levels are achieved. The variables used in the models are ∑GDD15 and ∑NHH15 for the day of achieving 15 °Brix DOY15, and ∑GDD20 and ∑NHH20 for the day of achieving 20 °Brix DOY20, respectively. 

The average values of accumulations over the 21 years for the 15 vineyards, were adopted as thresholds for the achievement of the two sugar contents. The performance of the models was tested by comparing the day of occurrence of the chosen thresholds with the observed day of occurrence of the sugar level. Statistical indices MAE, EF, and R^2^ [54] were used to assess the models’ performance. The obtained results are as follows:∑GDD15 = 1456∑NHH20 = 1763∑GDD20 = 1683∑NHH20 = 2037

To test the models’ performances in the calibration process, for each vineyard and each year, the day of occurrence of the thresholds was compared with the observed day of occurrence of sugar levels of 15 °Brix and 20 °Brix. The statistical indices MAE, EF, and R^2^ are reported in Table 8.

Regarding the estimation of DOY15 (Table 8), the NHH model shows better performances in terms of MAE, with an overall lower error of 8.03 days compared to the GDD model’s 10.17 days. Moreover, the NHH model provides better estimation values in 13 out of the 15 vineyards. However, the error is lower than one week only in five vineyards. Regarding EF, the NHH model performs better than GDD, but EF values are positive only for V1 and V15. This means that only for those two vineyards does the model provide a better estimation of DOY15 compared to the average of the observed DOY15 in the specific vineyard. In the case of R^2^, the GDD model shows better results in terms of significant regressions, but R^2^ values are always quite low.

The simulation of DOY20 showed worse but similar results. The global MAE increased to 10.40 and 14.56 for NHH and GDD, respectively. No vineyard shows MAE values below one week, and EF is positive only for vineyard V1.

However, analysing the averaged data for each vineyard (Figure 6), it is worth noting that the regressions are robust, with R^2^ values of 0.55 for GDD and 0.69 for NHH in the simulation of DOY 15. Similarly, for the estimation of DOY20, the regressions yield R2 values of 0.67 for GDD and 0.76 for NHH. 

To provide a long-term view of the ripening timing in Valtellina, the NHH models for DOY15 and DOY20 have been applied to the Sondrio time series of temperatures for the period 1981–2021 (Figure 7). Ten-year averages of the simulated DOY15 and DOY20 are also provided. Data from Vineyard V6 are shown for the period covered by monitoring activities to provide a qualitative idea of the model simulation. 

The simulation of DOY15 is more consistent than the simulation of DOY20 and shows weaker results for the first three years (2001, 2002, and 2003). 

Regarding the simulation of DOY 20, the model performances are particularly weak when ripening is late, as in the cases of 2006, 2008, and 2010. This could be explained by the fact that Nebbiolo ripens quite late in the season and the thermal-resources-based models are not able to accumulate enough resources at that time, especially during cold days. This confirms the idea that ripening models should not exclusively rely on temperature as the driving variable of the process. 

In more general terms, and focusing on the more reliable simulation of DOY15, the 10-year averages of the last four decades are 254 (1981–1990), 229 (1991–2000), 238 (2001–2010), and 236 (2011–2020), corresponding to 11 September, 17 August, 26 August, and 24 August. This confirms what was observed in the analysis of temperature: a warming phase in the 1990s that stabilised in the last 20 years.

## 3. Materials and Methods

### 3.1. Area of Study 

Valtellina is the most important longitudinal valley of the Lombardy Alpine chain. It is in the Northern part of the region and is crossed by the river Adda, one of the main tributaries of the river Po. Due to its east–west orientation, the valley is characterised by strong differences between the opposite slopes. On the southern side, the Orobian Alpine range faces north, and it is mostly characterised by woods and forests. On the northern side, the Rhaetian Alpine range faces south and, receiving strong sunlight, is traditionally devoted to agricultural activities, more specifically grape cultivation [43]. The morphology of this part of the slope is the result of the action of the glacier that occupied the valley until about 10,000 years ago. The slope is an alternation of areas where the emerging rocks have been smoothed and unhinged by the glacial exacerbation and areas covered by moraine and contact deposits. The natural morphology of the slopes has been modified over the centuries by important works of anthropic terracing (dry-stone wall terraces built between the 10th and the 14th centuries) that strongly characterise the landscape of the whole valley.

Viticulture has been mentioned in Valtellina since the Roman Empire, with references from Virgil and Pliny, who wrote about the quality of wine [55] and reached a peak of expansion in the mid-19th century with over 6000 hectares. The most widely cultivated grape variety is the red grape variety Nebbiolo, locally called Chiavennasca, which is fully adapted to the local conditions. Piedmont has traditionally been regarded as the ideal region for cultivating Nebbiolo, and outside of Piedmont, the cultivation of Nebbiolo vines has generally not yielded satisfactory results due to its specific microclimatic requirements [56]. However, in Valtellina, favourable conditions have been found for this grape variety, enabling the production of high-quality wines. The specific geographical features of the valley contribute to favorable growing conditions for grape cultivation in Valtellina. The valley’s configuration promotes consistent airflow, resulting in generally low relative humidity levels. Adequate and well-distributed precipitation occurs throughout the growing season. The vineyards benefit from high solar radiation input due to their optimal aspect. Additionally, the vineyards’ location on the middle belt of the slopes, away from the cold-air lake area at the bottom of the valley, facilitates a higher thermal gradient. 

The viticultural area covers a surface of about 1000 hectares, mostly located on the Rhaetian mountainside (southern slopes, and extends for about 70 km along an altitudinal belt comprised from the valley floor (200–300 m a.s.l.) up to about 700 m a.s.l. [57]. Wine production is regulated by the following appellations: the DOCG (Denominazione di Origine Controllata e Garantita—Controlled and Guaranteed Designation of Origin) “Sforzato di Valtellina” and “Valtellina Superiore,” the DOC (Denominazione di Origine Controllata—Controlled Designation of Origin) “Rosso di Valtellina,” and the IGT (Indicazione Geografica Tipica—Typical Geographical Indication) “Alpi Retiche.” The first three appellations are limited to the Rhaetian side along a 45 km belt centred on the main town of Sondrio, while “Alpi Retiche” covers a larger area, including vineyards located on the Orobian side of the valley and in the close north–south-oriented valley of Valchiavenna. 

### 3.2. The Vineyards

This study evaluates data collected by the technical assistance centre of Fondazione Fojanini in 15 commercial vineyards, representative of the viticultural belt along the Rhaetian mountainside of the valley (Figure 1 and Table 1). 

The vineyards’ technical characteristics are homogeneous: they sit on small artificial dry-stone terraces characterised by shallow, sandy, acid soil (pH between 4.5 and 5.5). The vines are trained to a modified Guyot system, locally known as Archetto Valtellinese, with a spacing of 1.7 m (within-row) × 1 m (in-row) and a row orientation along the maximum slope line. Additionally, 420A is used as a rootstock. The agronomic management of the vineyards is homogeneous, with within-row grass cover management and in-row mechanical management. Irrigation is not available due to difficult access to water. 

### 3.3. Grape Ripening Monitoring

Grape ripening in the 15 vineyards has been monitored since 2001, and the dataset analysed in this work covers the 2001–2021 period. From full veraison to harvest, 7 samplings (DOY 218, 228, 238, 248, 258, 268, and 278) were performed each year in each vineyard, for a total of 1575 records. During each sampling event, 100 berries were randomly selected from the vineyard. These berries were then collected for further analysis of their must composition. The timing of ripening was analysed considering the measurements of total soluble solids (TSS) and titratable acidity (TA). Total soluble solids (TSS) were determined on grapevine must by refractometry using a digital refractometer (DBR 35 SALT). Results were expressed as Brix (°Brix). pH and TA measurements were conducted using an automatic titrator. For each sample, 7.5 mL of juice was diluted to 50 mL with ultrapure water (resistivity > 18.2 MΩ·cm at 25 °C) to enable the analysis of TA. The samples were titrated with 0.1 M NaOH to a pH of 8.3 using an automatic titrator (FLASH Automatic Titrator, Steroglass). Acidity was expressed as g titratable acid L^−1^.

Based on the sampled data, for each vineyard and each season, a quadratic regression between sugar content and the sampling date [43], expressed as a day of the year (DOY), was set up in each vineyard to determine:DOY15—the day of achievement of a grape sugar content equal to 15 °Brix. This level has been chosen to represent an early ripening stage. This subjective threshold was chosen because it was the lowest value of the maturity curve that was always sampled in all the vineyards during the whole period of measurement, making the regression adopted for interpolation reliable.DOY20—the day of achievement of a grape sugar content equal to 20 °Brix, representing the technical maturity of grapes in Valtellina, that allows them to reach the minimum alcohol concentration (11% vol) requested by the local appellation (Valtellina DOCG).RL—the ripening length, as the period between DOY20 and DOY15.Furthermore, a quadratic regression between sugar content and total acidity was set up for each vineyard and each year to define:TA—the total acidity concurrent with grape technical maturity (20 °Brix).

### 3.4. Relationship between Ripening and Environmental Conditions

A general description of the climate of the study area was based on the analysis of data from the weather station located in Sondrio, the capital of the province, located in the middle of the viticultural area and close to the experimental vineyards V8, V9, V10, and V11.

To evaluate the ripening environmental conditions for the 15 vineyards, meteorological data for the period 2001–2021 were reconstructed. This data was obtained from a daily weather database that included information on maximum and minimum air temperature as well as cumulated precipitation. The database was sourced from the network of ARPA Lombardia, which is the Regional Environmental Agency. The network comprises 28 weather stations that cover the province of Sondrio and the northern part of the province of Lecco. The complete dataset of daily maximum and minimum temperatures and total daily precipitation was obtained for the 15 vineyards using spatialization, based on the geographical properties of each vineyard (coordinates, altitude, and elevation). The Digital Terrain Model (DTM) adopted for this work is released by the Lombardy Region [58] and has a resolution of 5 m x 5 m. The spatialization of temperature was based on inverse square distance weighting (IDW), taking into account elevation and aspect gradients [44,59,60]. Daily precipitation data were obtained with the IDW method. Hourly time series of temperature were obtained from the daily data using the Parton and Logan hourly generator [61].

The geographical and climatic features of each experimental vineyard were obtained by analysing the DTM and the meteorological data.

To evaluate the environmental resources and limitations during the ripening process, the following agrometeorological indices were calculated:

1. GDD—growing degree days with a 10 °C base [62]. This is a widely adopted method for the estimation of thermal resources useful for grapevine growth. Daily GDD is obtained from the daily average temperature Tmed as follows:If Tmed > 10 °C then GDD = Tmed − 10 else GDD = 0

Daily GDD are cumulated since 1 April.

2. NHH—normal heat hours [44,63]. This method of estimation of thermal resources overcomes the main limitation of GDD, which overestimates the effect of high temperatures on grape growth and does not consider the daily thermal ranges because it is based on the mean daily temperature. The normal heat hour approach applies a response function based on cardinal and optimal temperatures (Figure 2) to hourly temperatures. The parameterization of NHH for grapevine [64,65] assumes an optimal range of 24–26 °C and a cardinal range of 12–32 °C. With this approach, the hourly temperature is translated into a thermal effective hour: the score is zero when the temperature is outside the cardinal range and one when the temperature is within the optimal range. As temperature moves from the lower cardinal to the lower optimal, NHH linearly increases from zero to one, and when it moves from the upper optimal to the upper cardinal, NHH linearly decreases from one to zero. NHH are cumulated since the beginning of the year, starting when the moving ten-day average of Tmean overcomes 6 °C. 

3. HHH—high heat hour [44,63]. HHH is the complement to 1 of NHH for temperature values above the upper optimal. It represents the thermal stress conditions caused by over-optimal temperatures. HHH are cumulated since 1 January, starting when the moving ten-day average of Tmean overcomes 6 °C.

4. HW—refers to heat waves, which serve as an indicator of high-temperature stress. A day is considered a heat wave if the maximum temperature (Tmax) exceeds 32 °C.
If Tmax > 32 °C then HW = 1 else HW = 0 

HW are cumulated since 1 January.

5. Precip—the accumulation of precipitation since 1 January. Precipitation accumulation influences the timing of grapes and their ripening [28,51].

The calibration of predictive models for reaching 15 and 20 °Brix involved the following approach: for each vineyard and each season, the accumulations of NHH and GDD were computed until the day of achievement of the two sugar levels (∑GDD15, ∑NHH15 for DOY15 and ∑GDD20 and ∑NHH20 for DOY20, respectively). The average values of accumulations across the years were used as thresholds to determine the attainment of the two sugar contents. The performance of the models was tested by comparing the day of occurrence of the chosen thresholds with the observed day of occurrence of the sugar level. Statistical indices MAE, EF, and R^2^ [54] were used to assess the models’ performance.

## 4. Conclusions

Valtellina exhibited an increase in temperature that started in the 1990s and stabilised over the following 20 years. Inherently, during the last 21 years, the timing of ripening did not show significant trends.

As supported by other research on mountain viticulture, the timing of ripening (DOY15 and DOY20) and the total acidity (TA) were strongly correlated with altitude and temperature.

Thermal excess showed a negative correlation with the timing of technical maturity (DOY20) and total acidity (TA), but these factors did not have a detrimental impact on the quality of the wines. In fact, a few days with higher temperatures during ripening can be beneficial to ripening potential, and the current level of TA at harvest is optimal for the oenological goals of the area.

Precipitations showed good positive correlations with all the ripening indices, so that during more rainy years, later ripening and higher total acidity were observed.

The attempt to use thermal resources to predict the timing of ripening was not completely satisfactory. The model’s performance was not stable over the years, suggesting that other environmental variables played a relevant role. In this context, the good correlations obtained by precipitation in the description of the ripening dynamics strengthen this hypothesis.

While many viticultural areas have experienced climate changes that have posed challenges to grapevine growth, Alpine areas have benefited from environmental conditions that are conducive to high-quality production. These regions have observed earlier ripening of grapes, higher sugar levels, and well-maintained acidity, all of which contribute to the overall quality of the grapes. In this context, considering the current climate and the expected future scenarios, the use of higher-altitude vineyards could be a promising option for adaptation. In the specific context of Valtellina, this observation opens up the possibility of revitalising abandoned terraces situated at higher altitudes for viticultural activities. By utilising these marginal areas and harnessing the available environmental resources, valuable ecosystem services can be provided.

## Figures and Tables

**Figure 1 plants-12-02068-f001:**
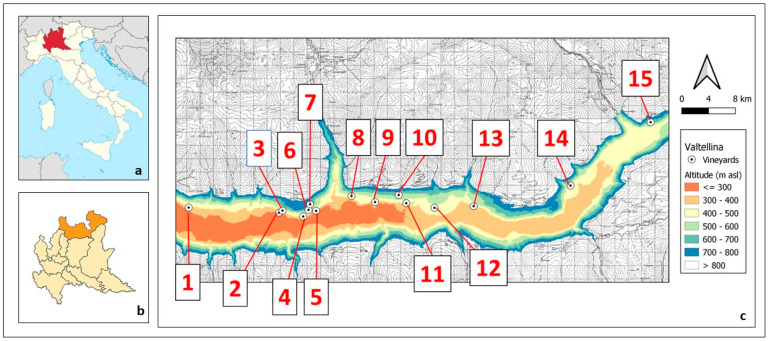
Location of the 15 vineyards and altitudinal map of Valtellina (**c**), an alpine valley located in the province of Sondrio (**b**) in the upper part of the Lombardy region (**a**), in the North of Italy.

**Figure 2 plants-12-02068-f002:**
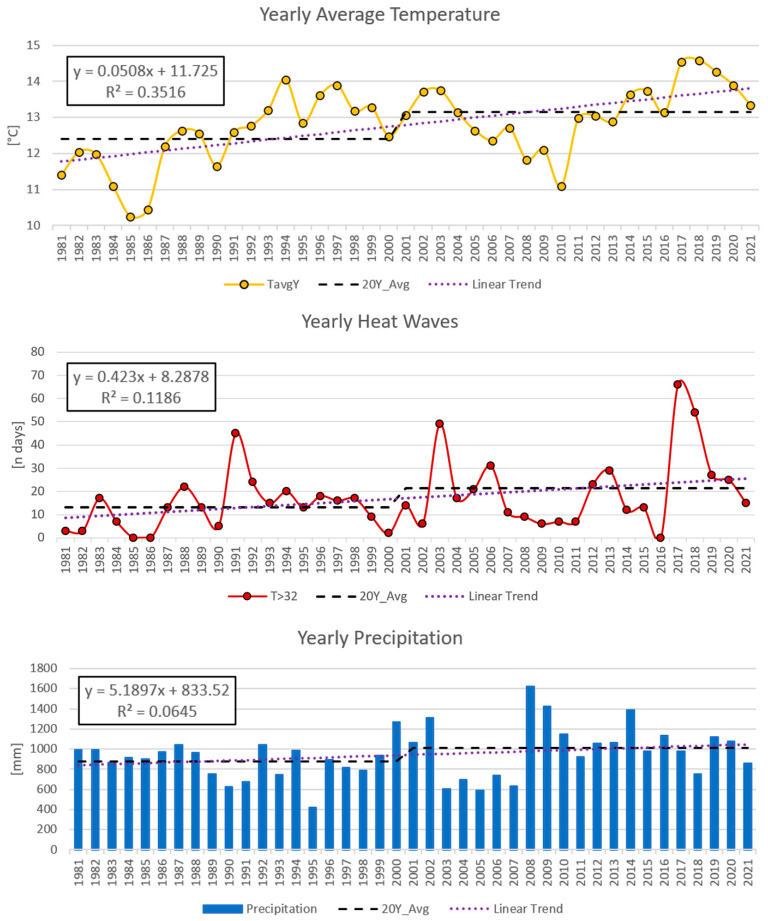
Sondrio (SO)—yearly values of average daily temperature, cumulated heatwaves (expressed as days with a maximum temperature above 32 °C), and cumulated precipitation. For each chart, the linear trend with R^2^ and the equation and the average value of the periods 1981–2000 and 2001–2020 are presented.

**Figure 3 plants-12-02068-f003:**
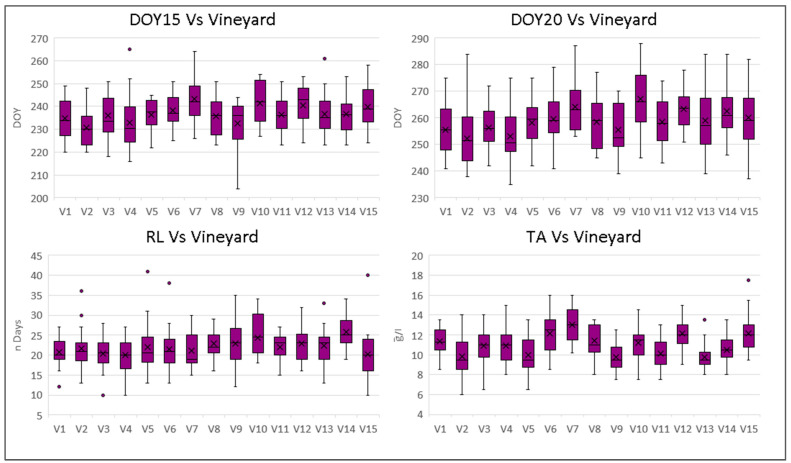
Statistics of the ripening indices DOY15, DOY20, RL, and TA. Each column represents the statistics of the index for the single vineyard (V1–V15), along with 21 years of data. For each column, “X” represents the average, the horizontal line is the median, and the box extends from the upper to the lower quartile. The whiskers (vertical lines outside the box) represent data variability outside the upper and lower quartiles. Points outside the whisker line represent outlier data.

**Figure 4 plants-12-02068-f004:**
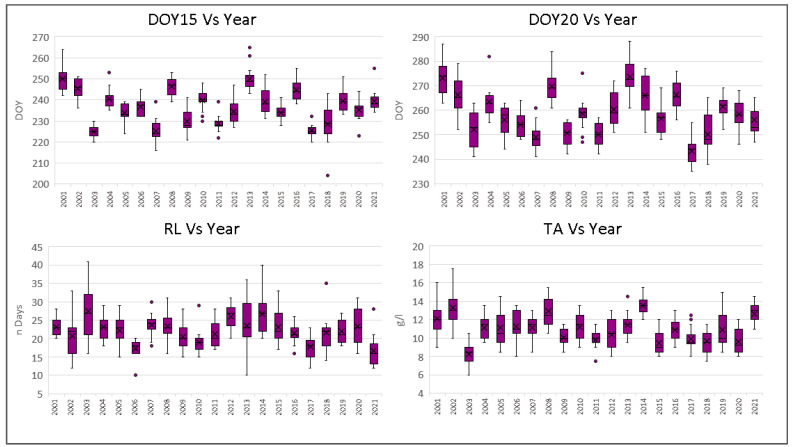
Statistics of the ripening indices DOY15, DOY20, RL, and TA. Each column represents the statistics of the index for a single year (2001–2021) among the 15 vineyards. For each column, “X” represents the average, the horizontal line is the median, and the box extends from the upper to the lower quartile. The whiskers (vertical lines outside the box) represent data variability outside the upper and lower quartiles. Points outside the whisker line represent outlier data.

**Figure 5 plants-12-02068-f005:**
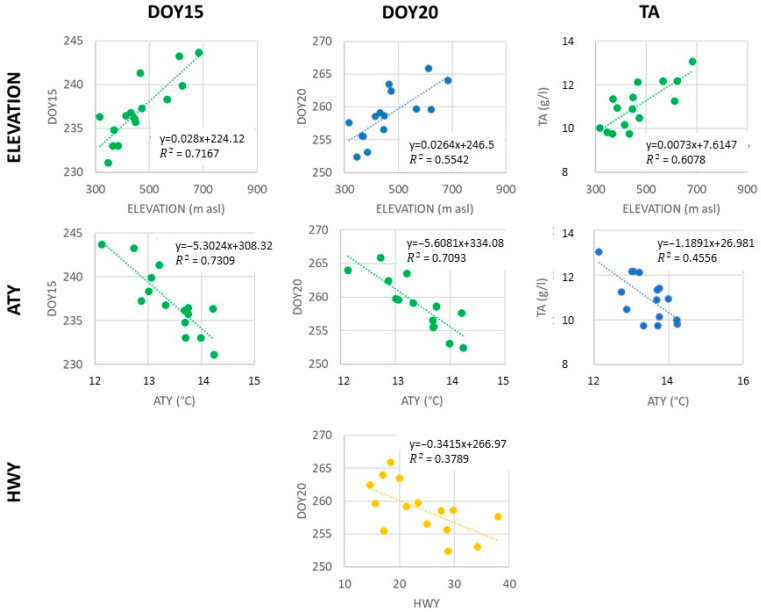
Linear regression between average values of DOY20, DOY15, RL, and TA for each vineyard and the geographical and climatic features (Elevation, ATY, and HWY). Colours represent different levels of significance of the regression (yellow = significant at 0.05 level; blue = significant at 0.01 level; green = significant at 0.001 level).

**Figure 6 plants-12-02068-f006:**
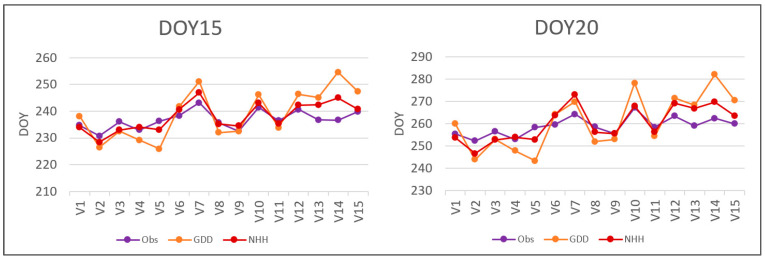
Vineyard averages over the 2001–2021 period of DOY15 and DOY20. Observed data (Obs) are compared with estimated data with the GDD model (GDD) and the NHH model (NHH).

**Figure 7 plants-12-02068-f007:**
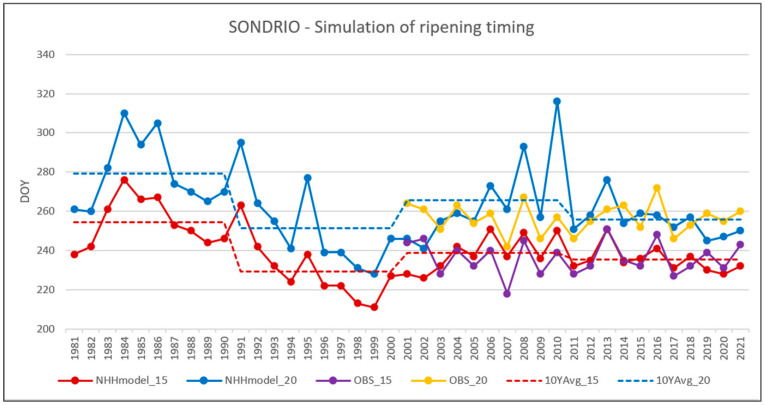
Long-time simulation of DOY15 and DOY20. NHH models (NHHmodel_15 and NHHmodel_20) are compared to observed data from vineyard 6 (OBS_15 and OBS_20). Ten-year averages of the simulated DOY15 and DOY20 are also provided (10YAvg_15 and 10YAVG_20).

**Table 1 plants-12-02068-t001:** Geographical and climatic features of the 15 vineyards.

Id	Site Name	X [°]	Y [°]	Elevation [m]	Aspect [°]	PPAR [MJ/m^2^]	ATY [°C]	HWY [n days]	PrecY [mm]
V1	Berbenno	9.72	46.17	370	175	3534	13.7	17	1244
V2	Castione	9.81	46.16	347	182	3565	14.2	29	1062
V3	Grigioni	9.81	46.16	446	190	3585	13.7	25	1025
V4	Triasso	9.83	46.16	386	160	3477	14	34	1023
V5	Sondrio	9.85	46.16	317	150	3476	14.2	38	1006
V6	Sant’Anna	9.84	46.16	568	157	3357	13	23	1022
V7	Triangia	9.84	46.17	685	135	3381	12.1	17	1006
V8	Montagna (Dossi)	9.88	46.17	449	172	3583	13.8	30	1039
V9	Montagna (Davaglione)	9.9	46.17	366	148	3444	13.7	29	1056
V10	Poggi Ridenti (Surana)	9.93	46.17	613	172	3701	12.7	18	1100
V11	Poggi Ridenti (Paradiso)	9.93	46.17	415	191	3527	13.8	28	1100
V12	Ponte	9.96	46.16	467	192	3317	13.2	20	1219
V13	Chiuro	10	46.16	433	187	3639	13.3	21	1255
V14	Bianzone	10.09	46.18	474	95	2794	12.9	15	1220
V15	Tirano	10.17	46.22	623	175	3592	13.1	16	1136
Average	-	-	-	464	165	3465	13	24	1101

**Table 2 plants-12-02068-t002:** Mean, maximum, and minimum values of the considered ripening indices.

Ripening Indices	Mean	Maximum	Minimum
DOY15 [doy]	237	254	221
DOY20 [doy]	259	287	235
RL [d]	22	33	14
TA [g L^−1^]	11.01	14.33	7.95

**Table 3 plants-12-02068-t003:** Regression between ripening indices and year. ** = significant at 0.01 level; * = significant at 0.05 level; ns = non-significant. In the case of significance, B—the angular coefficient of the regression equation—is reported; otherwise, the symbol is used. Negative relations are in red.

		DOY15			DOY20			RL			TA		
		R^2^	B	Sig.	R^2^	B	Sig.	R^2^	B	Sig.	R^2^	B	Sig.
	Total	0.09	-	ns	0.024	−0.345	**	0.01	−0.107	*	0.014	−0.037	*
Vineyard	V1	0.000	-	ns	0.057	-	ns	0.124	-	ns	0.037	-	ns
	V2	0.002	-	ns	0.067	-	ns	0.014	-	ns	0.041	-	ns
	V3	0.000	-	ns	0.002	-	ns	0.001	-	ns	0.030	-	ns
	V4	0.012	-	ns	0.028	-	ns	0.004	-	ns	0.129	-	ns
	V5	0.006	-	ns	0.053	-	ns	0.137	-	ns	0.000	-	ns
	V6	0.006	-	ns	0.148	−0.571	*	0.277	−0.464	**	0.007	-	ns
	V7	0.000	-	ns	0.015	-	ns	0.026	-	ns	0.081	-	ns
	V8	0.019	-	ns	0.015	-	ns	0.160	-	ns	0.066	-	ns
	V9	0.002	-	ns	0.001	-	ns	0.033	-	ns	0.000	-	ns
	V10	0.018	-	ns	0.187	−0.675	*	0.000	-	ns	0.072	-	ns
	V11	0.039	-	ns	0.123	-	ns	0.007	-	ns	0.045	-	ns
	V12	0.025	-	ns	0.001	-	ns	0.006	-	ns	0.001	-	ns
	V13	0.006	-	ns	0.187	−0.686	*	0.316	−0.401	**	0.028	-	ns
	V14	0.004	-	ns	0.074	-	ns	0.008	-	ns	0.091	-	ns
	V15	0.010	-	ns	0.059	-	ns	0.011	-	ns	0.166	-	ns

**Table 4 plants-12-02068-t004:** Regression results for DOY15 vs. HHH15, HW15, and Prec15. *** = significant at 0.001 level; ** = significant at 0.01 level; * = significant at 0.05 level; ns = non-significant. In the case of significance, B—the angular coefficient of the regression equation—is reported; otherwise, the symbol is used. Negative relations are in red; positive relations are in green.

		DOY15								
		HHH15			HW15			Prec15		
		R^2^	B	Sig	R^2^	B	Sig	R^2^	B	Sig
	Total	0.078	−0.019	***	0.089	−0.482	***	0.237	0.021	***
Vineyard id	V1	0.006	-	ns	0.006	-	ns	0.338	0.022	**
	V2	0.017	-	ns	0.021	-	ns	0.194	-	ns
	V3	0.039	-	ns	0.063	-	ns	0.184	-	ns
	V4	0.004	-	ns	0.002	-	ns	0.217	0.025	*
	V5	0.048	-	ns	0.022	-	ns	0.046		ns
	V6	0.183	-	ns	0.219	−0.208	**	0.374	0.022	**
	V7	0.207	-	ns	0.260	−0.301	*	0.322	0.027	*
	V8	0.151	-	ns	0.154	-	ns	0.288	0.022	*
	V9	0.046	-	ns	0.098	-	ns	0.273	0.025	*
	V10	0.178	-	ns	0.234	−0.336	*	0.194	-	ns
	V11	0.216	−0.031	*	0.241	−0.288	*	0.436	0.027	**
	V12	0.007	-	ns	0.036	-	ns	0.516	0.029	**
	V13	0.019	-	ns	0.037	-	ns	0.332	0.026	**
	V14	0.081	-	ns	0.125	-	ns	0.397	0.026	**
	V15	0.150	-	ns	0.212	−0.426	*	0.268	0.025	**

**Table 5 plants-12-02068-t005:** Regression results for DOY20 vs. HHH20, ΔHHH, HW20, ΔHW, Prec20, and ΔPrec. *** = significant at 0.001 level; ** = significant at 0.01 level; * = significant at 0.05 level; ns = non-significant. In the case of significance, B—the angular coefficient of the regression equation—is reported; otherwise, the symbol is used. Negative relations are in red; positive relations are in green.

		DOY20																	
		HHH20			ΔHHH			HW20			ΔHW			Prec20			Δprec		
		R^2^	B	Sig	R^2^	B	Sig	R^2^	B	Sig	R^2^	B	Sig	R^2^	B	Sig	R^2^	B	Sig
	Total	0.15	−0.03	***	0.36	−0.16	***	0.13	−0.22	***	0.25	−1.26	***	0.21	0.02	***	0.02	0.03	*
Vineyard id	V1	0.09	-	ns	0.48	−0.20	***	0.06	-	ns	0.27	−1.80	*	0.32	0.02	*	0.01	-	ns
	V2	0.02	-	ns	0.23	−0.13	*	0.02	-	ns	0.20	−1.05	*	0.17	-	ns	0.19	-	ns
	V3	0.16	-	ns	0.52	−0.18	***	0.16	-	ns	0.41	−1.56	**	0.18	-	ns	0.00	-	ns
	V4	0.15	-	ns	0.47	−0.19	***	0.11	-	ns	0.40	−1.57	**	0.20	0.02	*	0.00	-	ns
	V5	0.03	-	ns	0.12	-	ns	0.02	-	ns	0.14	-	ns	0.01	-	ns	0.10	-	ns
	V6	0.25	−0.03	*	0.26	−0.18	*	0.26	−0.25	*	0.17	-	ns	0.21	0.02	*	0.05	-	ns
	V7	0.22	-	ns	0.37	−0.29	*	0.22	-	ns	0.20	-	ns	0.28	0.02	*	0.00	-	ns
	V8	0.14	-	ns	0.39	−0.15	**	0.13	-	ns	0.33	−1.14	**	0.30	0.03	*	0.05	-	ns
	V9	0.18	-	ns	0.43	−0.09	**	0.19	-	ns	0.47	−0.88	***	0.28	0.02	*	0.08	-	ns
	V10	0.26	−0.04	*	0.63	−0.33	***	0.25	−0.37	*	0.35	−3.40	*	0.12	-	ns	0.00	-	ns
	V11	0.37	−0.04	**	0.50	−0.17	***	0.34	−0.30	**	0.39	−1.19	**	0.49	0.03	***	0.00	-	ns
	V12	0.00	-	ns	0.30	−0.16	*	0.02	-	ns	0.19	-	ns	0.32	0.02	*	0.00	-	ns
	V13	0.14	-	ns	0.44	−0.26	**	0.14	-	ns	0.23	−1.80	*	0.36	0.03	**	0.01	-	ns
	V14	0.09	-	ns	0.35	−0.23	**	0.03	-	ns	0.09	-	ns	0.20	0.02	*	0.00	-	ns
	V15	0.22	−0.05	*	0.31	−0.37	*	0.16	-	ns	0.02	-	ns	0.23	0.03	*	0.00	-	ns

**Table 6 plants-12-02068-t006:** Regression results for TA vs. HHH20, ΔHHH, HW20, ΔHW, Prec20, and ΔPrec. *** = significant at 0.001 level; ** = significant at 0.01 level; * = significant at 0.05 level; ns = non-significant. In the case of significance, B—the angular coefficient of the regression equation—is reported; otherwise, the symbol is used. Negative relations are in red; positive relations are in green.

		TA																	
		HHH20			ΔHHH			HW20			ΔHW			Prec20			Δprec		
		R^2^	B	Sig	R^2^	B	Sig	R^2^	B	Sig	R^2^	B	Sig	R^2^	B	Sig	R^2^	B	Sig
	Total	0.23	−0.01	***	0.34	−0.03	***	0.21	−0.05	***	0.51	−0.24	***	0.10	0.01	***	0.02	−0.01	**
Vineyard id	V1	0.25	−0.01	*	0.63	−0.03	***	0.19	-	ns	0.38	−0.31	**	0.11	-	ns	0.08	-	ns
	V2	0.31	−0.01	*	0.51	−0.03	***	0.33	−0.06	**	0.51	−0.28	***	0.18	-	ns	0.00	-	ns
	V3	0.25	−0.01	*	0.38	−0.03	**	0.28	−0.05	*	0.40	−0.36	**	0.34	0.01	**	0.00	-	ns
	V4	0.27	−0.01	*	0.37	−0.03	**	0.24	−0.04	*	0.35	−0.24	**	0.09	-	ns	0.06	-	ns
	V5	0.39	−0.01	**	0.53	−0.03	***	0.39	−0.05	**	0.53	−0.26	***	0.40	0.01	**	0.04	-	ns
	V6	0.04	-	ns	0.36	−0.05	**	0.05	-	ns	0.20	−0.34	*	0.18	-	ns	0.01	-	ns
	V7	0.28	−0.01	*	0.38	−0.06	*	0.28	−0.05	*	0.32	−0.81	*	0.10	-	ns	0.06	-	ns
	V8	0.17	-	ns	0.32	−0.02	**	0.18	-	ns	0.31	−0.18	**	0.08	-	ns	0.08	-	ns
	V9	0.27	0.00	*	0.23	−0.01	*	0.28	−0.03	*	0.22	−0.08	*	0.18	-	ns	0.09	-	ns
	V10	0.16	-	ns	0.31	−0.03	*	0.16	-	ns	0.22	-	ns	0.30	0.01	*	0.05	-	ns
	V11	0.05	-	ns	0.21	−0.02	*	0.05	-	ns	0.12	-	ns	0.10	-	ns	0.12	-	ns
	V12	0.21	-	ns	0.54	−0.04	***	0.25	−0.07	*	0.41	−0.34	**	0.15	-	ns	0.01	-	ns
	V13	0.18	-	ns	0.24	−0.02	*	0.16	-	ns	0.26	−0.22	*	0.03	-	ns	0.31	−0.02	**
	V14	0.60	-0.01	***	0.48	-0.04	***	0.47	-0.10	***	0.18	-	ns	0.43	0.00	**	0.15	-	ns
	V15	0.52	-0.01	***	0.42	-0.08	**	0.33	-0.10	**	0.10	-	ns	0.25	0.01	*	0.00	-	ns

**Table 7 plants-12-02068-t007:** Regression results for RL vs. ΔHHH, ΔHW, and ΔPrec. *** = significant at 0.001 level; ** = significant at 0.01 level; * = significant at 0.05 level; ns = non-significant. In the case of significance, B—the angular coefficient of the regression equation—is reported; otherwise, the symbol is used. Positive relations are in green.

		RL								
		ΔHHH			ΔHW			Δprec		
		R^2^	B	Sig	R^2^	B	Sig	R^2^	B	Sig
	Total	0.03	0.03	*	0.03	0.21	**	0.18	0.05	***
Vineyard id	V1	0.00	-	ns	0.00	-	ns	0.12	-	ns
	V2	0.02	-	ns	0.02	-	ns	0.26	0.06	*
	V3	0.05	-	ns	0.00	-	ns	0.21	0.04	*
	V4	0.09	-	ns	0.02	-	ns	0.44	0.05	***
	V5	0.22	0.07	*	0.18	-	ns	0.07	-	ns
	V6	0.06	-	ns	0.06	-	ns	0.07	-	ns
	V7	0.06	-	ns	0.09	-	ns	0.03	-	ns
	V8	0.03	-	ns	0.01	-	ns	0.35	0.05	**
	V9	0.25	0.04	*	0.15	-	ns	0.32	0.08	**
	V10	0.00	-	ns	0.00	-	ns	0.28	0.05	*
	V11	0.02	-	ns	0.01	-	ns	0.16	-	ns
	V12	0.23	0.07	*	0.28	0.70	*	0.14	-	ns
	V13	0.00	-	ns	0.00	-	ns	0.09	-	ns
	V14	0.02	-	ns	0.05	-	ns	0.19	-	ns
	V15	0.00	-	ns	0.03	-	ns	0.13	-	ns

**Table 8 plants-12-02068-t008:** Performance of the ripening timing models based on GDD and NHH, MAE, EF, and R^2^ (with significance) is reported. *** = significant at 0.001 level; ** = significant at 0.01 level; * = significant at 0.05 level; ns = non-significant.

	DOY15								DOY20							
	MAE		EF		R^2^		Sig.		MAE		EF		R^2^		Sig.	
	GDD	NHH	GDD	NHH	GDD	NHH	GDD	NHH	GDD	NHH	GDD	NHH	GDD	NHH	GDD	NHH
All	10.17	8.03	−1.01	−0.35	0.28	0.13	***	***	14.56	10.40	−2.64	−0.76	0.18	0.11	***	***
V1	7.70	5.70	−0.14	0.30	0.37	0.34	**	**	11.00	6.80	−2.72	0.19	0.20	0.31	ns	*
V2	8.15	6.15	−0.66	−0.07	0.20	0.15	*	ns	12.90	9.50	−1.36	−0.27	0.08	0.12	ns	ns
V3	8.52	6.52	−0.87	−0.28	0.20	0.12	*	ns	12.86	8.71	−3.41	−1.23	0.20	0.03	*	ns
V4	9.60	9.60	−0.38	−0.13	0.13	0.07	ns	ns	12.05	10.70	−1.28	−0.73	0.15	0.01	ns	ns
V5	12.00	8.40	−3.58	−1.31	0.14	0.00	ns	ns	16.70	9.95	−6.09	−1.67	0.04	0.01	ns	ns
V6	10.52	7.81	−2.04	−0.82	0.27	0.04	*	ns	14.37	11.60	−2.92	−2.12	0.19	0.01	ns	ns
V7	14.63	10.88	−2.22	−0.99	0.16	0.00	ns	ns	21.71	14.20	−16.91	−1.90	0.27	0.01	ns	ns
V8	8.90	8.81	−0.77	−0.95	0.20	0.00	*	ns	15.14	11.48	−2.29	−1.38	0.09	0.01	ns	ns
V9	7.95	9.25	−0.07	−0.52	0.28	0.00	*	ns	12.40	10.95	−1.82	−1.12	0.21	0.00	*	ns
V10	9.82	8.71	−0.85	−0.19	0.29	0.08	*	ns	17.94	11.19	−4.32	−0.49	0.12	0.08	ns	ns
V11	6.43	6.81	−0.11	−0.34	0.38	0.05	**	ns	10.43	8.86	−1.18	−0.90	0.34	0.01	**	ns
V12	9.83	7.67	−1.28	−0.28	0.19	0.07	ns	ns	14.06	10.94	−4.42	−3.30	0.03	0.05	ns	ns
V13	10.81	9.00	−1.06	−0.32	0.27	0.16	*	ns	12.95	11.25	−0.90	−0.51	0.33	0.21	*	*
V14	18.75	10.75	−6.25	−1.39	0.36	0.06	**	ns	25.56	11.67	−13.75	−2.06	0.01	0.00	ns	
V15	10.00	4.90	−0.95	0.54	0.57	0.55	***	***	16.00	9.37	−2.91	−0.40	0.25	0.25	ns	*

## Data Availability

Data available at Alpine Viticulture and Climate Change.

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
