# Peer review of "Alpine Viticulture and Climate Change: Environmental Resources and Limitations for Grapevine Ripening in Valtellina, Italy"

_plants, 2023, doi:10.3390/plants12112068_

Round 1

Reviewer 1 Report

The ms plants-2396842 with the title of Alpine viticulture and Climate Change: environmental resources and limitations for grapevine ripening in Valtellina (Italy) investigates a very interesting topic but the authors should improve their ms before it can go for further process.

Title, please remove the dot at the end of the title and change title to be: ……. in Valtellina, Italy

Affiliations should be added

L12-16 please reduce this text to two lines as background.

L16 The aim of this work WAS not is …

L37-47 merge these two paragraphs together as well as please make sentences shorter to make them easier for readers. Sentences are very long!

L58-66 merge these two paragraphs together as well as please make sentences shorter to make them easier for readers. Sentences are very long!

L67-68 please cite this ref: https://doi.org/10.1016/j.agwat.2020.106626

L67-85 merge these paragraphs together.

L129 evaluated NOT evaluates, please change the text to past tense when you write about something done in the past.

Results and discussion: The discussion is superficial and the authors should improve it as well as they should improve the quality of the figures, particularly x and y axes

The colour of the text in Figures should be BLACK, this can be done easily in Excel where you created the Figures.

Table 1: The column of X [°], the authors should use two or three decimals for the values but do not make mix, please.

Table 2: Can you make any statistical analysis for such data since you have different values for ripening indices?

Figures 3, 4 and others: what are the bars indicates in each column? Please add this in the text under each figure.

Table 3: what does mean by -? Does it mean NO significant? Or what? If yes, then use ns, no significant

Some Figures and Tables can be moves as supplementary materials

Conclusions: The authors should focus on what they want to say for the readers and remove other common text. Please reduce this section to 50% percent.

The authors should improve their ms.

Author Response

We thank the reviewer for the helpful work done. Here attached the file with the answers to both the reviewers. In the revised manuscript we left in red the changed parts.

Reviewer 2 Report

The study discusses the influence of climate on viticulture, specifically in mountainous wine regions like Valtellina in Italy. This study found that altitude, temperature, summer thermal excess, and precipitation all play a role in grape ripening and total acidity levels. Overall, the study provides valuable insights for viticulturists and wineries in mountainous wine regions on how to optimize grape ripening in response to changing climatic conditions.

Some minor issues should be addressed before publication.

1. Too many paragraphs in the introduction. Authors should try to mere them and make it more concise.

2. Line 89-90. I think you were talking about waterlogging sensitivity. This is also very common in many crops such as barley. Authors could cite this reference to further support the statement.

Liu, K., Harrison, M.T., Yan, H. et al. Silver lining to a climate crisis in multiple prospects for alleviating crop waterlogging under future climates. Nat Commun 14, 765 (2023). https://doi.org/10.1038/s41467-023-36129-4

3. Why some numbers in Table 3-7 are in red or green?

4. Conclusion section is too long and not concise. Please consider revising it.

English needs to be checked by native speakers.

Author Response

(The authors gave the same response as above.)

Round 2

Reviewer 1 Report

ms has been improveed

Author Response

Thank you for the work done

As suggested by the editor we removed references from the section 5

We moved some of them in the introduction. In the main text, changes are in red
